# Extract of Tangerine Peel as a Botanical Insecticide Candidate for Smallholder Potato Cultivation

**DOI:** 10.3390/insects16070680

**Published:** 2025-06-29

**Authors:** José-Manuel Pais-Chanfrau, Lisbeth J. Quiñonez-Montaño, Jimmy Núñez-Pérez, Julia K. Prado-Beltrán, Magali Cañarejo-Antamba, Jhomaira L. Burbano-García, Andrea J. Chiliquinga-Quispe, Hortensia M. Rodríguez Cabrera

**Affiliations:** 1School of Agro-Industrial Engineering, Universidad Técnica del Norte, Ibarra 100105, Imbabura, Ecuador; 2School of Agricultural Engineering, Universidad Técnica del Norte, Ibarra 100105, Imbabura, Ecuador; 3School of Biotechnology Engineering, Universidad Técnica del Norte, Ibarra 100105, Imbabura, Ecuador; 4Yachay Tech Medicinal Chemistry Research Group (MEDCHEM-YT), School of Chemical Sciences and Engineering, Yachay Tech University, Urcuquí 100115, Imbabura, Ecuador; hmrodriguez@yachaytech.edu.ec

**Keywords:** botanical insecticide, *Citrus reticulata* L., *Solanum tuberosum* L., Citrus peel ethanolic extract, beneficial insects

## Abstract

The overuse of synthetic pesticides in agriculture poses substantial risks to human health, beneficial insects, and the environment, while driving pesticide resistance in pests. This study explored an eco-friendly alternative by developing ethanolic extracts from tangerine (*Citrus reticulata* L.) peels as a botanical pesticide for small-scale potato (*Solanum tuberosum* L.) farming. Two formulations (1.25% and 2.50%, *v*/*v*) were tested against conventional chemical treatments and an untreated control during the 2023 wet season in Ecuador. Field results demonstrated that the 2.50% formulation reduced pest populations (e.g., *Frankliniella occidentalis*, Aphididae) by 49–62%, matching the efficacy of synthetic pesticides, while preserving beneficial insects such as ladybirds and honeybees. Post-harvest analysis revealed that the botanical treatments achieved 73% of conventional yields, with comparable tuber quality and reduced infestations by *Premnotrypes vorax* larvae. Toxicological tests confirmed the extract safety, showing minimal harm to non-target organisms such as *Caenorhabditis elegans*. This research valorises agro-industrial waste and offers a sustainable pest management solution for organic farming and honey production in Andean *chakras*, aligning with circular bioeconomy principles. Further refinements could bridge yield gaps, promoting global adoption of greener agricultural practices.

## 1. Introduction

The large-scale use of chemical substances in agriculture to maintain soil fertility and prevent the harmful effects of pests and diseases on major crops has contributed to the high yields and food security that modern agriculture demonstrates today [1]. Initially, sufficient food supplies were achieved, at least in terms of production, to nourish the growing global population [2]. However, the uncontrolled and excessive use of chemical agents compromises the long-term sustainability of agriculture and global food security [3,4].

Although highly effective, some chemical substances may also cause various health disorders in humans and the surrounding flora and fauna. Additionally, the effectiveness of some pesticides has declined over time as the pests they target have mutated and acquired various resistance genes, allowing them to survive contact with these agents [5].

Imidacloprid, a widely used insecticide, effectively controls many insects, most of which are agricultural pests. However, studies have shown that workers who handle this chemical insecticide are at serious risk of health issues due to prolonged exposure [6]. It has also been linked to a drastic decline in beneficial insects, such as the honeybee [7,8]. Promoting awareness and implementing measures are critical steps towards minimising these risks in the future.

While botanical pesticides may not be as potent for stopping pests that harm crops, they show promising results for the future of agriculture [9,10,11]. They have the advantage of having a lower impact on human health and a less adverse effect on agroecosystems. In some cases, they have been incorporated into integrated pest management programs aiming to reduce the negative impact of agrochemicals without significantly affecting current agricultural yields [12,13].

One example of botanical insecticides that has gained attention involves the use of essential oils from citrus peels [14,15]. These peels are relatively rich in monoterpenes, such as limonene and linalool, which have demonstrated insecticidal properties [15].

In a previous study, essential oil from tangerine peels (*Citrus reticulata* L.) effectively controlled whiteflies (*Trialeurodes vaporariorum* Westwood) [16]. These extracts were obtained using nonpolar organic solvents (petroleum ether and n-hexane), the toxicity and flammability of which make them challenging to use in broader contexts [16,17].

The current study uses ethyl alcohol as a polar solvent to extract bioactive compounds from the tangerine peel (*Citrus reticulata* L.). This choice of solvent is much less toxic, flammable, and safer than petroleum ether and n-hexane, providing reassurance about the safety of the research methods [18].

Other authors have also reported that essential oils obtained from *C. reticulata* possesses antimicrobial activity [19,20,21,22,23], including some insect-transmitted infections, such as ‘cucumber wilt,’ a serious illness caused by the bacterium *Erwinia tracheiphila* [24].

Additionally, alcoholic extracts of *C. reticulata* peel have been found to contain some flavonoids, especially poly-methoxy-flavonoids, such as tangeretin and nobiletin, which are attributed to antifungal properties against certain fungal pests such as *Botrytis cinerea*, *Sclerotinia sclerotiorum*, and *Fusarium oxysporum* [24,25].

In general, many extracts and essential oils from the peels of other members of the *Citrus* genus have also demonstrated potential in controlling pathogenic bacteria [26,27,28,29,30,31] and fungi [32,33,34,35]. For example, *C. limon* and *C. sinensis* have shown effectiveness against potato rot disease [33].

This study used potatoes (*Solanum tuberosum* L.) as a model crop. The current research was conducted in a small field plot under controlled conditions, using two levels of ethanolic tangerine extract.

This research aims to evaluate two formulations as potential botanical pesticide substitutes for conventional chemical treatments in controlling fungal and insect pests during the rainy months in Ecuador. This is the period when these pests are most prevalent.

## 2. Materials and Methods

### 2.1. Raw Materials

*Citrus reticulata* L. var. Clementine was obtained from the Pimampiro Canton (0°24′0″ N, 77°58′12″ W), situated in the northern part of Ecuador. The tangerine fruits were harvested in January 2023. The fruits collected reached maximum ripeness when the skin easily detached from the pulp [17].

Before peeling, the tangerines were cleaned with abundant tap water at room temperature to remove any possible dust from shipping. Each fruit was dried using a clean and dry cloth. Following the peeling, the peels were cut into small pieces (cubes of 1–2 cm) to optimise ethanolic extraction, as suggested by previous studies [16,17]. The cut peels were weighed before being subjected to extraction with ethyl alcohol.

### 2.2. Pilot-Scale Extraction–Concentration of Ethanolic Extract of Tangerine Peels

The ethanol in the tangerine peel extract was removed, and the extract was concentrated in a pilot unit for discontinuous solvent extraction, coupled with a condenser for solvent separation (Armfield Limited, Hampshire, England, BH24 1DY, UK).

The pilot unit for solvent extraction utilises saturated steam at 3 barg, and the condenser employs cooling water at 10–12 °C to condense and recirculate the solvent (ethanol). It also features a pump that recirculates the condensed solvent through a distributor onto the tangerine peels. The pump is activated when a specific level is reached in the collection tank for the condensed solvent, and six such cycles were used throughout the extraction process (Figure 1).

The fresh, washed, and cut tangerine peel weighed 9.87 kg. The material was mixed with 20 litres of 96% ethyl alcohol, technical grade (for a solvent-to-solid ratio of approximately 2.03 L kg^−1^), and macerated for 15 h.

Subsequently, the ethanolic extract began to evaporate, condense, and recirculate for about 5–6 h (six whole cycles) in the pilot unit. Afterwards, the alcohol was evaporated and separated from the extract for 2 h, ensuring that the condenser temperature remained below 90 °C. The operation was halted when the temperature rose above 90 °C, concluding the material concentration process and the safe removal of all the solvent.

The material was stored in a 1-litre amber bottle at 4 °C until its use in field experiments and analyses.

### 2.3. Characterisation of the Ethanolic Extract of Tangerine Peels

#### 2.3.1. Phytochemical Screening of Ethanolic Tangerine Peel Extract

The ethanolic extract was subjected to phytochemical analysis using standard procedures reported in the literature, with minor modifications [36,37,38,39].

##### Protein and Amino Acids Determinations (Ninhydrin Assay)

A mixture of equal volumes of tangerine peel ethanolic extract and 0.2% (*m*/*v*) ninhydrin solution (freshly prepared) in a test tube was heated for 1–2 min, resulting in a blue-to-dark purple product, demonstrating the existence of amino acids and proteins [36].

##### Detection for Phenols and Tannins

In a test tube, equal amounts (1 mL) of ethanolic extract and 2% (*m*/*v*) FeCl_3_ solution were mixed to give a blue-green or black colouration, indicating that phenols and tannins are present [36].

##### Test for Carbohydrates

An equal volume of ethanolic extract was combined with Benedict’s reagent in a test tube and heated to a boiling point (approximately 100 °C) for 3–5 min. The emergence of a reddish-brown precipitate signifies the presence of carbohydrates [37].

##### Flavonoid Assays (Alkaline Reagent and Shinoda Tests)

An equal volume of ethanolic extract was combined with a 2% NaOH solution in a test tube, resulting in a pronounced yellow colouration. The presence of flavonoids results in a colourless solution upon the addition of several drops of concentrated hydrochloric acid [38].

Additionally, the Shinoda test was employed to detect the presence of flavonoids. In a test tube, 1 mL of ethanolic extract and 10 drops of diluted hydrochloric acid were mixed, followed by the addition of 500 mg of magnesium. The presence of flavonoids results in the production of a reddish, pink, or brown colouration [39].

##### Test for Steroids (Libermann Test)

In a test tube, 1 mL of ethanolic extract was combined with 2 mL of chloroform and 2 mL of acetic acid. Then, the mixture was cooled for 10 min by placing the sample on ice, and then 0.5 mL of concentrated sulfuric acid was slowly added at intervals. If, after that, the colour turns into violet, blue, or green, this implies the presence of steroidal molecules [38].

#### 2.3.2. FTIR Study of Ethanolic Tangerine Peel Extract

The ethanolic extract from tangerine peels was analysed using Fourier-transform infrared (FTIR) spectrometry (Agilent Cary 630 FTIR, Agilent Technologies Inc., Santa Clara, CA, USA) across a wavenumber range of 400 to 4000 cm^−1^, with 32 scans performed at a resolution of 4 cm^−1^. Additionally, an attenuated total reflection (ATR) sampling method was employed to analyse a single-rebound diamond crystal [16].

The FTIR spectrum of tangerine peel extract can help identify the main chemical functional groups present in the extract’s components.

#### 2.3.3. Reversed-Phase HPLC

The samples were analysed using an RP-HPLC system (Ultimate 3000 HPLC, equipped with a reverse-phase C-18 column (150 × 4.6 mm) from HPLC Hypersil GOLD, an autosampler, a Thermo Scientific quaternary pump (Waltham, MA, USA), a column compartment, and a photodiode array detector (PAD). The chromatograms were produced at a wavelength of 205 nm using a linear gradient of H_2_O/CH_3_CN (95:5) to (40:60) for 8 min [16].

Through the RP-HPLC spectrum obtained from the extract of tangerine, it would be possible to identify the presence and composition within the mixture of some of the main substances that constitute the extract.

#### 2.3.4. Atmospheric Pressure Gas Chromatography–Mass Spectrometry (APGC-MS)

APGC-MS detection was performed using a 7890A gas chromatograph outfitted with a SYNAPT G2 HDMS detector, using an Agilent DB5-MS column, 30 m in length, with a 0.25 mm inner diameter and a 0.25 mm film thickness (5% phenyl and 95% polydimethylsiloxane). A split injection ratio of 100:1 and an injection temperature of 230 °C were used, with helium serving as the carrier gas.

The temperature was 70 °C, increasing at a rate of 5 °C/min to 300 °C. The transfer line temperature was 310 °C.

Mass spectrometry was conducted on the SYNAPT G2 HDMS (Waters Corporation, Milford, MA, USA) utilising TOF-MSE (positive ion mode) and APCI ionisation. The corona current and sampling cone voltage were 2.8 mA and 30 V, respectively. A source temperature of 150 °C, a cone gas flow rate of 50 L h^−1^, and an auxiliary gas flow rate of 500 L h^−1^ were utilised.

### 2.4. Toxicity Evaluation Using the Caenorhabditis Elegans Model

#### 2.4.1. *Caenorhabditis elegans* Strain Culture

*C. elegans* wild-type N2-strain worms were obtained from the Caenorhabditis Genetics Centre (CGC) in collaboration with the EvoDevo Laboratory, Bioscience Institute of São Paulo University.

*C. elegans* wild-type N2-strain worms were cultured on NGM agar plates using *Escherichia coli* strain OP50 as feed. The nematode culture was maintained at 20 °C and subcultured every two weeks [40]. Each culture was synchronised by bleaching with a sodium hypochlorite solution treatment and cultured for two days at 20 °C until the L2 nematode stage [41,42].

#### 2.4.2. Toxicity Assay and LC_50_ Estimation

Toxicity assays were performed in 96-well culture plates in a final volume of 200 μL of buffer S solution (0.1 M K_2_HPO_4_, 0.1 M KH_2_PO_4_, 0.1 M NaCl, at pH 7.0, with 1 mL L^−1^ of adjuvant ARPON (polyether-polymethyl-siloxane copolymer) with 1, 6, 12.5, 25, and 75% (*v*/*v*) of tangerine peel ethanolic extract (Eth-E)) [40].

Twenty L2 nematodes were transferred into each well. Four wells were prepared for each dose, and this assay was conducted in triplicate.

Ivermectin (6.0 mg mL^−1^) was used as a positive control. Buffer S solution with 1 mL L^−1^ of adjuvant ARPON (polyether-polymethyl-siloxane copolymer) was used as the negative control, and 1 mL L^−1^ of polyether-polymethyl-siloxane copolymer (ARPON) solution was used as the solvent control [43].

Plates were incubated at 20 °C for 24, 48, and 72 h. After exposure, the live and dead worms were counted by visual inspection under a dissection microscope. Worms were considered dead if they did not respond to light or mechanical stimuli.

Results were presented as the mean survival percentage of three replicates. The statistical significance was evaluated by one-way ANOVA followed by Tukey’s test. Finally, the lethal concentration (LC_50_) was calculated for the PROBIT analysis [44,45].

### 2.5. Field Experiments on Potato (Solanum tuberosum L.) Cultivation

The field experiments were conducted between January and April of 2023 on the grounds of the experimental farm ‘La Pradera,’ located at the Universidad Técnica del Norte and situated in the canton of Chaltura, province of Imbabura, Ecuador (0°22′31.3″ N; 78°21′20.5″ W).

The ‘Capiro’ variety of *Solanum tuberosum* L. was used for the small-scale potato cultivation experiment. A completely randomised block design of experiments was used. Two aqueous extracts—1.25% (*v*/*v*), prepared by mixing 12.5 mL of extract and 1 mL of adjuvant ARPON (polyether-polymethyl-siloxane copolymer) and tap water to 1 L; and 2.50% (*v*/*v*), prepared by mixing 25 mL of extract + 1 mL of adjuvant, and tap water up to 1 L—together with conventional chemical management (‘conv. treat.’), and an ‘untreated’ variant (in which only received fertilisation but without any application of chemical or botanical pesticide) were carried out, with three blocks per treatment.

The maximum extract dose selected for the experiments of 2.50% (*v*/*v*) is related to some phytotoxicity observed with aqueous extract concentrations of 3.00% (*v*/*v*) in the leaves of *S. tuberosum*.

Each of the twelve experimental units (3 blocks × 4 treatments) consisted of three rows of 15 potato plants per row, with distances of 100 cm between rows and 40 cm between plants within rows (Figure 2).

Each experimental unit had an area of 10.80 m^2^ and was 300 cm away from neighbouring experimental units. The order of treatments was randomised in each of the three blocks (Figure 2). Yellow traps were placed at the centre of each experimental unit (Figure 2).

The dynamics of the various insect groups in the agroecosystem were closely observed, and the yields obtained from the potatoes were compared.

Ethanolic extracts, formulated at 1.25% (*v*/*v*) and 2.50% (*v*/*v*) (F-1.25% and F-2.50%), were applied every seven days. The total volume used for each treatment, F-1.25% and F-2.50%, was approximately 1 L per treatment and week. The applications were carried out between 7:30 and 9:30 a.m. One mL L^−1^ of polyether-polymethyl-siloxane copolymer (ARPON) was added as an adjuvant to promote the formation of a homogeneous emulsion. Adjuvant was also included in the conventional chemical treatment.

Each treatment was supplemented with fertilisers during the experiment, as described in Table 1.

Direct (through the detailed visual inspection of each plant and counting eggs, nymphs, aphids, and caterpillars) and indirect (through recording the different species of insects caught in the yellow traps) monitoring were implemented to measure the variables. In indirect monitoring, the numbers of eggs and nymphs of *Bactericera cockerelli* S., aphids, and Lepidoptera larvae were counted on 15 plants of each treatment. Each plant was divided into three parts, and in each part, the number of these insects was counted on three leaves.

The percentage reduction was determined relative to the quantity detected in the ‘untreated’ variant and was calculated as: [(quantity observed ‘untreated’ − quantity observed in another treatment)/quantity observed ‘untreated’] × 100, assuming that the most observed individuals will be in the ‘untreated’ variant.

Yellow traps (10 cm × 25 cm) were used to monitor the units. These were changed every 15 days, and the numbers of *F. occidentalis*, *B. cockerelli*, *Epitrix* spp., aphids, and adult leaf miners were recorded.

### 2.6. Post-Harvest Analysis

The yields of each treatment block (in ton ha^−1^) were recorded. Additionally, a random sample of 10 potato tubers per block was collected from each of the three blocks within each treatment (each treatment consisted of three similar blocks) to assess visible external damage, allowing for the determination of the percentage of damaged tubers. The damaged tubers were then carefully cut open to count the white larvae they contained [46].

### 2.7. Statistical Analysis of Experiments

The statistical language R (RStudio 2024.12.0+467) was used for statistical analysis and the creation of some graphs.

The Shapiro and Bartlett tests were used to evaluate the assumptions of normality and the homogeneity of variance. If the samples were normally distributed and their variances homogeneous, an analysis of variance (ANOVA) was employed; otherwise, the Kruskal–Wallis test was used. In addition, if these tests (ANOVA or Kruskal–Wallis) yielded *p*-values < 0.05, a post hoc pairwise comparison test must be employed using the parametric Tukey or non-parametric Friedman or Dunn tests.

## 3. Results

### 3.1. Pilot-Scale Solid–Liquid Extraction

In the extraction–concentration process, 992 mL of tangerine peel ethanolic extract (with a dark brown colour and characteristic smell) and a density of 985 g L^−1^ was obtained from 9.87 kg of initial tangerine peels, extracted with 20 litres of 96% ethanol. The overall yield was: [(0.992 L × 0.985 kg L^−1^)/9.87 kg] × 100 = 9.90% (*w*/*w*).

### 3.2. Characterisation of Ethanolic Extract from Tangerine Peels

Phytochemical screening of the tangerine peel ethanolic extract was conducted using various chemical assays to identify the presence of secondary metabolites, including phenols, tannins, flavonoids, proteins, amino acids, reducing sugars, saponins, carbohydrates, steroids, terpenoids, and alkaloids (Table 2).

The FTIR spectrum exhibited peaks indicative of primary interactions among the atoms related to the extract composition (Figure 3).

The initial peak, located at around 3426 cm^−1^, is indicative of the stretching vibration of the O-H bond, which is likely attributable to the presence of pectin, a substance abundant in citrus peels. The 3000 to 2800 cm^−1^ peaks correspond to the asymmetric stretching of aliphatic C-H bonds in ethyl (-CH_3_) and methylene (-CH_2_) groups. The pronounced peak at 1627 cm^−1^ indicates C=C bond stretching, likely associated with the cyclohexene limonene ring. Peaks at roughly 1442 cm^−1^ indicate the doublet stretching of the C-H bond in the methylene groups. These results are similar to those obtained in another study, which used tangerines of the exact origin and extracted tangerine peel extracts with polar solvents such as n-hexane and petroleum ether [16].

As a result, the aqueous extract of tangerine peel contained functional groups commonly found in secondary metabolites, such as terpenes (e.g., limonene), phenols, flavonoids, and steroids [16].

The HPLC detected a few compounds that were susceptible to UV-Vis analysis in the aqueous tangerine peel extract (Figure 4a).

The HPLC chromatogram (Figure 4a) displayed a single prominent peak in the sample at a retention time of 2.113 min, accounting for ~70% of the total detected compounds in the entire sample.

The peak at a retention time of 2.113 min was collected and analysed using an APGC-MS system to confirm the presence of *d*-limonene. The mass spectrum of the collected sample showed a prominent peak corresponding to the molecular ion of limonene (*m*/*z* 136.0560) (Figure 4b). A similar finding was reported for a sample of the same origin, but extracted with n-hexane [16].

### 3.3. Toxicity Evaluation of Ethanolic Extracts of Tangerine Peels on Caenorhabditis elegans Model

The results demonstrated a dose-dependent effect of the ethanolic extract (suspended in ARPON) on the mortality of L2-nematodes after 24 h of exposure, as illustrated in Figure 5a. The PROBIT analysis yielded an LC50 value of 6.43 ± 0.28% (n = 3). 

The data indicate that the tangerine peel ethanol extract exhibits a significant anthelmintic effect in *C. elegans* models. It produces substantial mortality, ranging from 25% to 75%, after 24 h of testing, and this progressively increases over time.

Qualitative microscopic analysis of the nematodes’ digestive integrity revealed notable disruptions in their structure and functionality. These findings suggest that the extract and the solvent resulted in a direct acute toxic effect on the nematodes’ digestive system and cuticle.

Interestingly, the solvent used in the assay reduced nematode survival compared to the control (-) with solvent (*p* = 6.88 × 10^−5^). Some evidence suggests that the silicone in ARPON (polymethyl siloxane copolymer) can cause inflammation and irritation in the long term [33], which justifies the effect on the cuticle observed in *C. elegans* (Figure 5b). However, the combined use of tangerine ethanolic extract shows a protective effect, improving nematode survival (Eth-E-1% and Eth-E-6%).

### 3.4. Field Experiments

The application with the formulations resulted in more *Bactericera cockerelli* S. eggs and nymphs than the other pests. On the other hand, the predominant species among the aphids was *Macrosiphum euphorbiae* T., while among the caterpillars, *Spodoptera frugiperda* J.E.S. was the most prevalent.

Additionally, in the treatment with the aqueous extract formulation at 1.25% (*v*/*v*) (F-1.25%) and the untreated variant (hereafter referred to as ‘untreated’), a greater quantity was observed compared to the other treatments throughout the entire experiment.

Direct observations of the number of eggs, nymphs, aphids, and caterpillars revealed that these values changed significantly with increased cultivation time and varied according to the treatment received, as indicated by the Kruskal–Wallis test (*p* < 0.05) (Figure 6).

For instance, for the eggs (Figure 6a), this average reduction was 21%, 49%, and 57% with the F-1.25%, F-2.50%, and conventional chemical treatment (‘conv. treat.’), respectively. In the case of the nymphs (Figure 6b), the reduction was 25%, 52%, and 61% in the same order. For the aphids (Figure 6c), the reduction was 64%, 52%, and 75%, while for the caterpillars (Figure 6d), the reduction was 50%, 62%, and 100%.

In general, the number of eggs, nymphs, aphids, and caterpillars significantly decreased over time between 30 and 90 days of cultivation (*p* < 0.05, Friedman test).

A similar significant decrease and difference (*p* < 0.05, Friedman test) were observed in the order ‘untreated’ → ‘F-1.25%’ → ‘F-2.50%’ → ‘conv. treat.’ for the number of eggs, nymphs, and caterpillars.

However, in the case of aphids, the order of significant decrease is ‘untreated’ → ‘F-2.50%’ → ‘F-1.25%’ → ‘conv. treat.’. There is no convincing explanation for the behaviour of the aphids (Figure 6c), in which the F-1.25% treatment was more statistically (*p* < 0.05, Friedman test) effective than the F-2.50% treatment. Complementary experiments will be conducted to confirm this unexpected observation.

On the other hand, the number of insects of each class counted in the yellow traps exhibited a non-normal distribution (Figure 7 and Figure 8).

The insect pest with the highest presence in the yellow traps was thrips, which showed a lower population with the ‘conv. treat’. In some weeks, however, it was identical to the application of F-2.50% (Figure 7a).

Similar behaviour was observed for the aphids on days 75 and 90 when applying ‘conv. treat.’, F-1.25%, and F-2.50%, while for the last count, a lower population was noted with F-2.50% and ‘conv. treat.’, except on day 60, where the values for the conventional chemical treatment, F-1.25%, and ‘untreated’ were similar. Similarly, the population of *Bactericera cockerelli* was reduced by F-2.50% and ‘conv. treat.’ in the last three measurements (Figure 7b).

For *Epitrix* spp., the most significant effect on its population was achieved with the applications of F-2.50% and the ‘conv. treat.’ after 90 planting days. Applying F-2.50% resulted in counts approximately 5–6 insects lower than for F-1.25% and conventional treatment, while counts were approximately 13–15 lower than in the ‘untreated’ control (Figure 7c).

The results show that in the plots with conventional chemical treatment (‘conv. treat.’), fewer than three individuals of ladybugs (Figure 8a) and wasps (Figure 8b) were present, and honeybees (*Apis mellifera*) (Figure 8c) were nearly absent from this treatment. However, the extracts’ application did not affect the populations of beneficial insects, with the F-2.50% and ‘untreated’ doses showing eight more specimens of Hymenoptera compared to the F-1.25% applications, except for the planting times of 75 and 90 days, where the ‘untreated’ treatment was significantly different to the rest of treatments (Figure 8b).

The presence of beneficial insects and pollinators is practically absent from the yellow traps in the blocks where conventional chemical treatment was applied (Figure 8).

The presence of these insects in the treatments with the ethanolic extracts (F-1.25% and F-2.50%) and the ‘untreated’ variant was statistically similar (*p*-value > 0.05) for most of the evaluated times (Figure 8).

### 3.5. Post-Harvest Analysis

Conventional management reached ~36 tons ha^−1^ for the highest yielding unit, surpassing the experimental units F-2.50% and F-1.25% by ~10 tons ha^−1^. The plots without applications yielded 24 tons ha^−1^, demonstrating the damage caused by different pests on yield (Figure 9).

Finally, the selected tuber samples from each treatment block (n = 10) were thoroughly examined and cut. The number of damaged tubers was recorded, and then the number of white grub larvae (*Premnotrypes vorax* Hustache, Coleoptera: Curculionidae) present in each damaged tuber was counted, allowing for the calculation of the average number of white grub larvae per treatment (Figure 10).

The ‘untreated’ variant was consistently observed to have significantly higher values (*p* < 0.05) than the other treatments. Additionally, no differences were observed between the conventional treatments, specifically F-1.25% and F-2.50%, in terms of the average value of damaged tubers.

However, concerning the average number of counted white grub larvae, no differences were observed between the conventional treatment and the F-2.50% treatment, nor between the F-2.50% and F-1.25% treatments. However, differences were noted between the latter and the conventional chemical treatment (Figure 10).

This result suggests that the treatments studied, based on aqueous extracts of mandarin peels, can extend their protective action to harmful soil insects, such as *P. vorax*, the Andean potato weevil (Figure 10a). However, their number in the damaged tubers does not differ statistically (*p* > 0.05, Tukey test) from the treatment that did not use any pesticides (Figure 10b).

## 4. Discussion

The present study substantiates the potential of aqueous extracts from tangerine (*Citrus reticulata* L. var. ‘Clementina’) peels as a sustainable botanical insecticide. Phytochemical screening, corroborated by Fourier-Transformed Infrared (FTIR), Reversed-Phase High-Performance Liquid Chromatography (RP-HPLC), and APGC-MS analyses, confirmed the presence of a rich profile of secondary metabolites, notably phenols, flavonoids, and steroids, with a predominant (−70%) *d*-limonene component. These findings are consistent with previous characterisations of citrus peel extracts [16,47] and form the biochemical basis for the observed biological activities. The complexity of such natural extracts, often containing a synergistic blend of compounds, is a hallmark of botanical pesticides and may contribute to a broader spectrum of activity and a reduced likelihood of rapid pesticide resistance development compared to single-molecule synthetic pesticides [13,48,49].

The insecticidal efficacy of the 2.50% (*v*/*v*) formulation (F-2.50%) was particularly noteworthy, achieving reductions in foliar pest populations, such as *Frankliniella occidentalis* and Aphididae, by 49–62%. This level of control approached that of the conventional chemical treatments, a significant finding for a plant-derived product. The insecticidal properties of citrus extracts, particularly *d*-limonene, are well-documented, often attributed to neurotoxic effects, desiccation of the insect cuticle, or disruption of octopaminergic pathways [13,50,51,52,53,54,55]. Furthermore, the observed reduction in *Premnotrypes vorax* larvae infestation in tubers suggests a broader efficacy that extends to soil-dwelling pests, an area requiring further investigation but highly relevant for potato cultivation, where such pests cause significant economic losses [46,56,57]. The antifungal activity associated with flavonoids in citrus extracts [21,24,25,58] may also have contributed to overall plant health, although specific assessments of fungal diseases were beyond the scope of this study.

A critical advantage highlighted by this research is the F-2.50% formulation’s minimal impact on beneficial insect populations, including Coccinellidae and *Apis mellifera*. This contrasts sharply with the conventional chemical treatment, which demonstrably suppressed these natural enemies and pollinators. The selectivity of many botanical insecticides, which allows for the conservation of beneficial arthropods, is a cornerstone of integrated pest management (IPM) strategies [13,59,60,61]. This attribute makes the tangerine peel extract particularly suitable for agroecosystems, such as the Andean ‘chakras’, where biodiversity and ecosystem services, including pollination for organic honey production [10,62,63,64], are highly valued. The low toxicity, further evidenced by the *Caenorhabditis elegans* model, where field-relevant concentrations (approximating the Eth-E-1% sample) showed minimal adverse effects, underscores the formulation’s eco-friendly profile, likely due to the near-complete removal of ethyl alcohol during the pilot-scale concentration process.

Whilst the F-2.50% formulation yielded approximately 73% of the tuber production achieved with conventional chemical management, representing a difference of roughly 10 ton ha^−1^, this outcome must be contextualised within a broader sustainability and circular bioeconomy framework [65,66]. Several factors could partially offset the slightly lower yield. Firstly, the reduced reliance on synthetic pesticides mitigates environmental burdens, such as soil and water contamination, and slows the development of pesticide resistance in pest populations [67]. Secondly, the valorisation of agro-industrial waste (tangerine peels) aligns with biorefinery concepts, potentially creating additional revenue streams or reducing waste disposal costs, thereby improving the overall economic equation for farmers [68,69,70]. Thirdly, produce grown with such botanical inputs may command premium prices in organic or eco-labelled markets, compensating for yield differentials [71]. The long-term benefits of enhanced soil health and preserved biodiversity, although not directly quantified in this study, also contribute to the resilience and sustainability of the farming system.

The promising results advocate integrating tangerine peel extracts into pest management programs, particularly for smallholder farmers seeking sustainable and ecologically sound agricultural practices. Future research should focus on optimising extraction efficiency and field application protocols to bridge the yield gap alongside a more comprehensive quantitative analysis of the extract’s chemical constituents and their specific modes of action. Investigating the extract’s efficacy during different climatic conditions, such as the ‘dry’ season, will also be crucial for validating its robustness across varied environmental contexts.

## 5. Conclusions

This study successfully characterised the ethanolic extract derived from *Citrus reticulata* L. var. ‘Clementina’ peels contain bioactive secondary metabolites, including phenols, flavonoids, and a significant proportion (−70%) of *d*-limonene. Field trials revealed that the 2.50% formulation achieved pest control efficacy comparable to conventional chemical treatments, reducing populations of *Frankliniella occidentalis*, Aphididae, and *Premnotrypes vorax* larvae by 49–62%, while preserving beneficial insect populations, such as Coccinellidae and *Apis mellifera*. Toxicological assessments on *Caenorhabditis elegans* indicated minimal environmental toxicity at field-relevant concentrations, reinforcing the formulation’s safety. Although tuber yields under botanical treatments reached 73% of conventional yields, the absence of harmful agrochemical residues and compatibility with organic farming practices highlight their suitability for Andean ‘chakras’, where sustainable agriculture and honey production are prioritised. This approach aligns with circular bioeconomy principles by valorising agro-industrial waste, offering a viable strategy to reduce chemical dependency and enhance ecological resilience in smallholder systems. Further refinement of extraction efficiency and field application protocols may bridge yield gaps, advancing the integration of botanical pesticides into global pest management frameworks.

## Figures and Tables

**Figure 1 insects-16-00680-f001:**
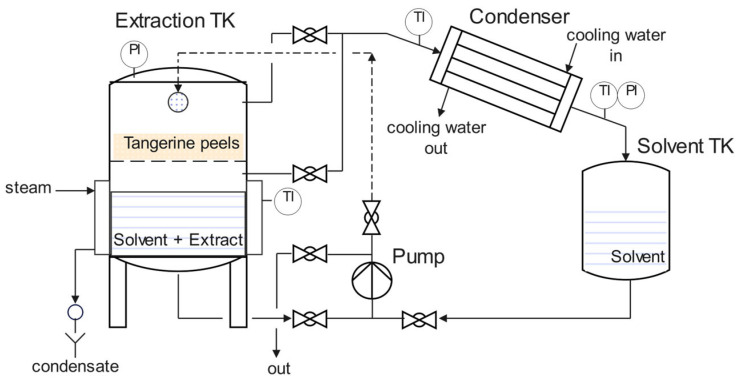
Diagram of the pilot extractor–concentrator.

**Figure 2 insects-16-00680-f002:**
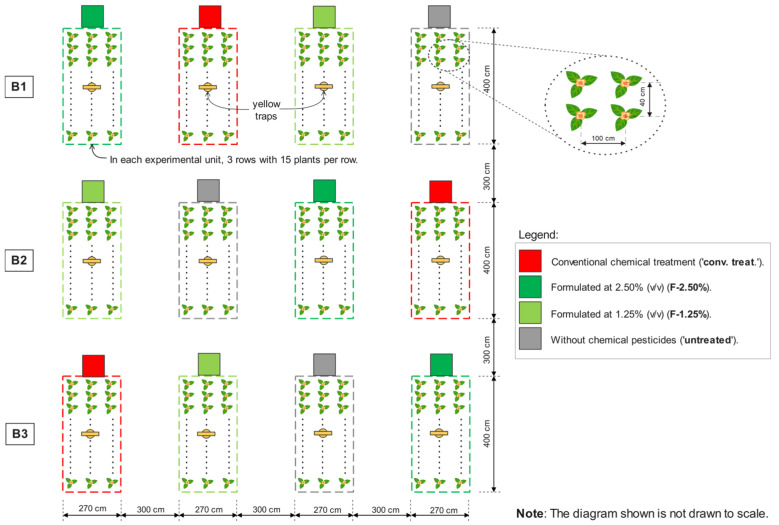
Diagram of the field experiment conducted with potato crops (*Solanum tuberosum* L.). In each of the three experimental blocks (B1–B3), the four experimental treatments were randomly distributed and tested.

**Figure 3 insects-16-00680-f003:**
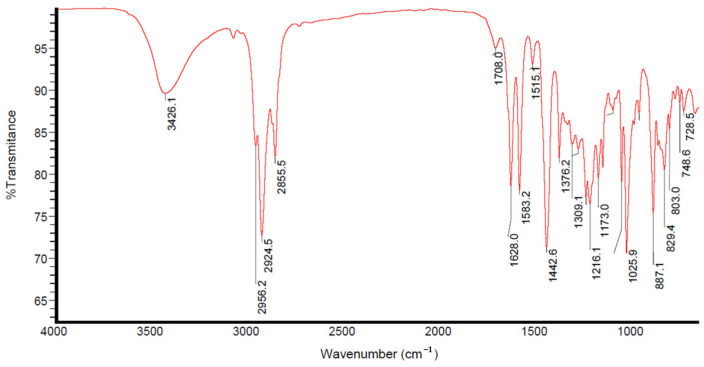
Tangerine peel aqueous extract FTIR scanning between 4000−400 cm^−1^.

**Figure 4 insects-16-00680-f004:**
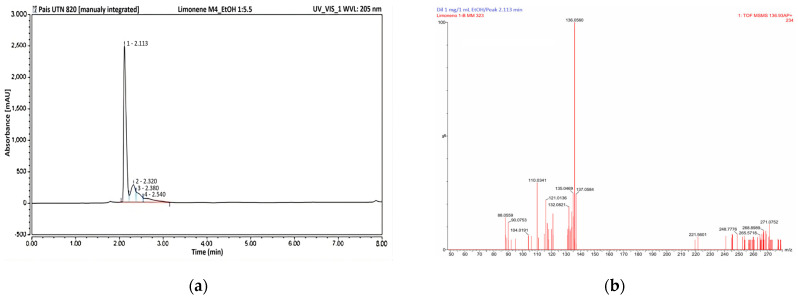
Characterisation of the prominent peaks of RP-HPLC. (**a**) RP-HPLC profile of ethanolic extract of tangerine peels at 205 nm; (**b**) APGC-MS detection of the RP-HPLC peak of 2.113 min.

**Figure 5 insects-16-00680-f005:**
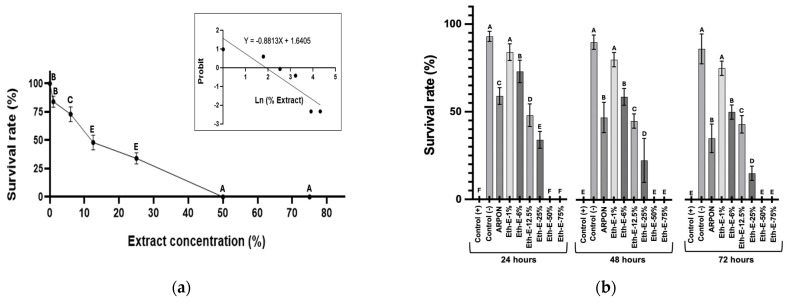
Toxicological effects of the ethanolic extract of tangerine peels on *Caenorhabditis elegans*. (**a**) Concentration–effect curve of nematodes exposed to the extract for 24 h. Each point represents the average of three experiments, with standard deviation values shown under one-way ANOVA and the Tukey test. The Probit regression analysis yields an *R*^2^ value of 0.8044 and a *p*-value of 0.0154. (**b**) Survival of *C. elegans* in different extract concentrations at different experimental times, analysed by one-way ANOVA with Tukey test (α = 0.05). Different letters denote statistical differences (*p*-value < 0.05) each time.

**Figure 6 insects-16-00680-f006:**
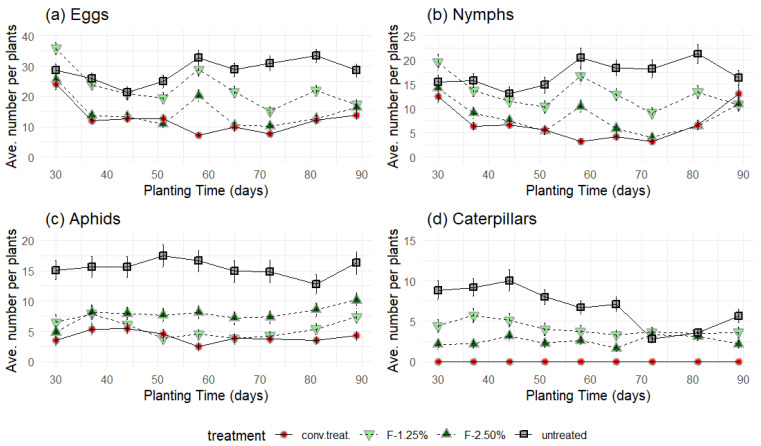
Direct count of (**a**) eggs, (**b**) nymphs, (**c**) aphids, and (**d**) caterpillars observed before each treatment. The value represents the average of three treatment blocks, and the error bars represent the standard deviation.

**Figure 7 insects-16-00680-f007:**
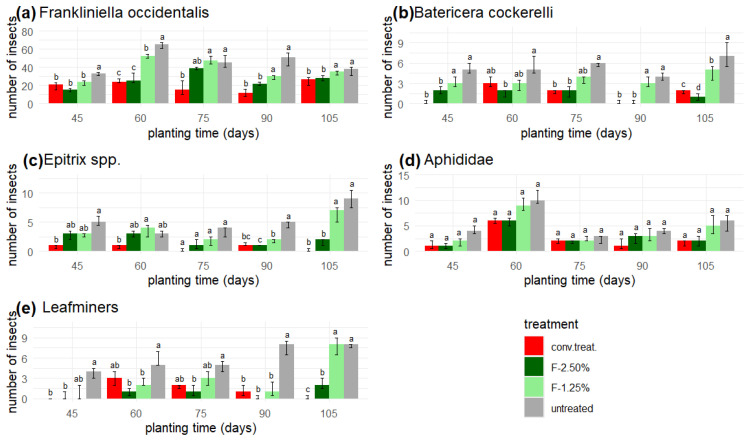
Yellow traps: Insect pests. (**a**) *Franklinella occidentalis*, (**b**) *Bactericera cockerelli*, (**c**) *Epitrix* spp., (**d**) Aphididae, (**e**) Leafminers in yellow traps: insect pests. The bar above each column represents the interquartile range, and the unequal letters indicate statistically significant differences in median values (*p* < 0.05) within each planting time, as determined by the Friedman test.

**Figure 8 insects-16-00680-f008:**
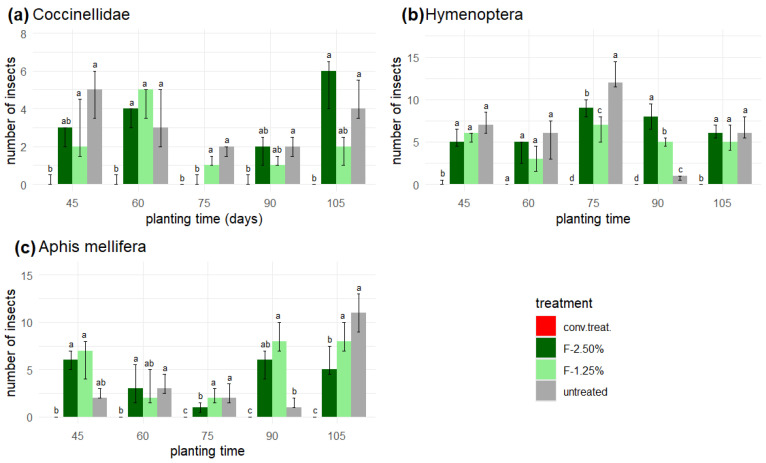
Yellow traps: beneficial insects or natural enemies. (**a**) Coccinellidae, (**b**) Hymenoptera (except *Apis mellifera*), and (**c**) *Apis mellifera*. According to the Friedman test, the presence of identical letters within each planting time indicates no significant differences (*p* > 0.05).

**Figure 9 insects-16-00680-f009:**
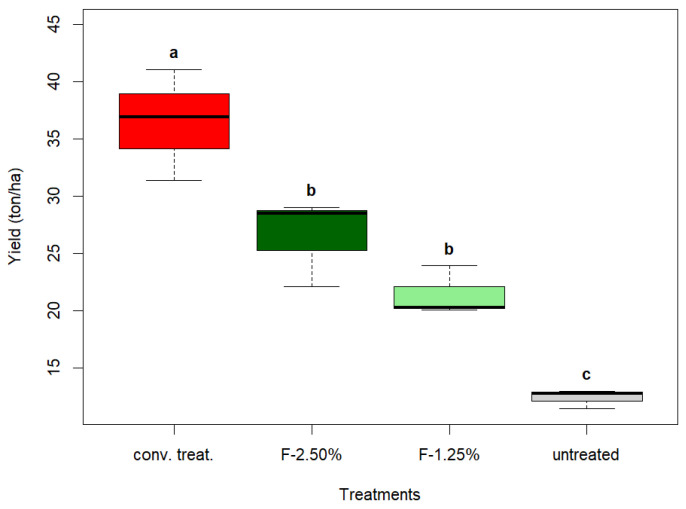
The yield of potatoes per treatment. The values represent the values of three blocks of experiments. According to the Tukey test, the letters denote statistically significant differences (*p* < 0.05).

**Figure 10 insects-16-00680-f010:**
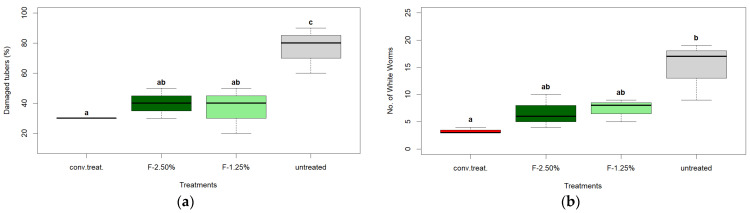
Post-harvest analysis of samples (n = 10 per block and three blocks per treatment) of the different treatments. (**a**) Percentage of damaged tubers; (**b**) Average white grub larvae (*P. vorax* H.) detected in the damaged tubers. According to the Tukey test, the unequal letters indicate significant differences (*p* < 0.05).

**Table 1 insects-16-00680-t001:** Fertilisation scheme used in all treatments and the chemicals employed in the conventional chemical treatment (‘*conv. treat.*’). Symbol: Applied (’+’), Not applied (’−’).

Name	Function	‘*conv. treat.*’	F-2.50%	F-1.25%	‘*untreated*’	Dose	Frequency	Mode of Action ^‡^
‘Biol’	organic fert.	**+**	**+**	**+**	**+**	5 L ^†^	Every month	-
Compost	organic fert.	**+**	**+**	**+**	**+**	300 kg ^†^	Start of cultivation	-
Calcium carbonate	Soil pH adjust.	**+**	**+**	**+**	**+**	50 kg ^†^	Start of cultivation	-
NPK (13-40-13)	inorganic fert.	**+**	**+**	**+**	**+**	7.46 kg ^†^	Every two months	-
Imidacloprid	Insecticide	**+**	−	−	−	0.5 mL·L^−1^	Every two weeks	(i)
Thiamethoxam + lambda cyhalothrin	Insecticide	**+**	−	−	−	125 mL·L^−1^		(ii)
Acephate	Insecticide	**+**	−	−	−	100 g·L^−1^		(iii)
Fipronil	Insecticide	**+**	−	−	−	2 mL·L^−1^		(iv)
Methomyl	Insecticide	**+**	−	−	−	3 mL·L^−1^		(iii)
Malathion	Insecticide	**+**	−	−	−	2 mL·L^−1^		(iii)
Fluopicolide + Propamocarb chlorhydrate	Fungicide	**+**	−	−	−	1.6 mL·L^−1^	Every two weeks	(v)
Propineb + Fluopicolide	Fungicide	**+**	−	−	−	2 mL·L^−1^		(vi)
Carboxin + Captan	Fungicide	**+**	−	−	−	3 g·L^−1^		(vii)

^†^ These values represent the total amounts applied throughout the experiment. The value shown can be divided into four to determine the amount used in each treatment. Additionally, it can be divided by the total area of the three blocks for each treatment (3 × 2.70 m × 4.0 m = 32.40 m^2^) to express it per unit area in each treatment. ^‡^ To view the rankings, please check the following link: https://irac-online.org/mode-of-action/classification-online/ (accessed on 24 June 2025): (i) Nicotinic acetylcholine receptor competitive modulators (nAChR); (ii) Sodium channel modulators + Receptor (nAChR) competitive modulators; (iii) Acetylcholinesterase (AChE) inhibitors; (iv) GABA-gated chloride channel blockers; (v) Disrupting the function of spectrin-like proteins in the fungal cytoskeleton + targeting their phospholipid biosynthesis; (vi) Multiple metabolic processes + disrupting the function of spectrin-like proteins in the fungal cytoskeleton; (vii) Disrupting fungal respiration + hindering their energy production.

**Table 2 insects-16-00680-t002:** Qualitative phytochemical analysis of tangerine peel ethanolic extract.

Test	Tangerine Peel Ethanolic Extract ^†^
Amino Acids and Proteins	−
Phenols and Tannins	+
Carbohydrates	−
Flavonoids (Alkaline test)	+
Flavonoids (Shinoda test)	+
Steroids	+

^†^ Detected (‘+’) and not detected (‘−’).

## Data Availability

The original contributions presented in this study are included in the article. Further inquiries can be directed to the corresponding author.

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
