# Peer review of "Extract of Tangerine Peel as a Botanical Insecticide Candidate for Smallholder Potato Cultivation"

_insects, 2025, doi:10.3390/insects16070680_

Round 1

Reviewer 1 Report

Comments and Suggestions for Authors

The manuscript evaluates the potential of an ethanolic extract from tangerine peel as an insecticide for controlling potato pests. The subject is relevant, and the experimental design is appropriate, including field evaluations. I recommend accepting the manuscript for publication; however, several major issues need to be addressed by the authors:

  • Lines 391–393: This comparison should be evaluated statistically, as it is unclear whether the differences observed are significant.
  • Lines 462–464: Revise this sentence, as the figure suggests the absence of significant differences between the untreated and conventional methods (see Fig. 9ab).
  • Discussion section: This section needs a thorough revision. It largely repeats the results, including figures (see lines 468–480, 483–491, and 508–509), and some parts are merely a review of the literature or general comments (lines 481–482, 492–494, 510–520).
  • Additionally, relevant findings regarding insect control, potato yield, and damage should be discussed in more depth.

Minor:

  • Line 45: Replace Aphis with Apis.
  • In the Methods section, clarify what was evaluated using FTIR and HPLC, as many readers may not be familiar with these techniques.
  • Explain how the extract concentrations were applied in the field.
  • Lines 378–380 and 395–397: These sentences should be moved to the Methods section.

Author Response

Dear Reviewer 1,

I hope you are doing well. Thank you very much for your thorough and detailed review of our manuscript, which will significantly enhance its quality.

Answer to Reviewer 1:

Comments 1: Lines 391–393: This comparison should be evaluated statistically, as it is unclear whether the differences observed are significant.

Respond 1: I agree with you. Indeed, this part was poorly explained. To address this comment and further clarify this point, lines 385-387 and 393-402 have been inserted in the new version of the manuscript (highlighted in yellow).

Lines 385-387:

Direct observations of the number of eggs, nymphs, aphids, and caterpillars revealed that these values changed significantly with increased cultivation time and varied according to the treatment received, as indicated by the Kruskal-Wallis test (p < 0.05) (Figure 6).

Lines 393-402:

In general, the number of eggs, nymphs, aphids, and caterpillars significantly decreased over time between 30 and 90 days of cultivation (p< 0.05, Friedman test).

A similar significant decrease and difference (p < 0.05, Friedman test) were observed in the order 'untreated' -> 'F-1.25%' -> 'F-2.50%' -> 'conv. treat.' for the number of eggs, nymphs, and caterpillars.

However, in the case of aphids, the order of significant decrease is 'untreated' -> 'F-2.50%' -> 'F-1.25%' -> 'conv. treat.'. There is no convincing explanation for the behaviour of the aphids (Figure 6c), in which the F-1.25% treatment was more statistically (p < 0.05, Friedman test) effective than the F-2.50% treatment. Complementary experiments will be conducted to confirm this unexpected observation.

Comments 2: Lines 462–464: Revise this sentence, as the figure suggests the absence of significant differences between the untreated and conventional methods (see Fig. 9ab).

Respond 2: To clarify this point, changes have been inserted in lines 473, 475-477.

Line 473:

Changed “ethanolic” to “aqueous” for more precision.

Ethanol was used solely for the extraction of secondary metabolites from tangerine peels and was removed before use. The extract was formulated with tap water for potato experiments in the field, so the term “aqueous” is more precise than “ethanolic”.

Lines 475-477:

…(Figure 10a). However, their number in the damaged tubers does not differ statistically (p > 0.05, Tukey test) from the treatment that did not use any pesticides (Figure 10b).

Comments 3: Discussion section: This section needs a thorough revision. It largely repeats the results, including figures (see lines 468–480, 483–491, and 508–509), and some parts are merely a review of the literature or general comments (lines 481–482, 492–494, 510–520). Additionally, relevant findings regarding insect control, potato yield, and damage should be discussed in more depth.

Respond 3: To clarify this point, the 'Discussion' section has been completely rewritten in lines 479-537.

Lines 479-537:

The present study substantiates the potential of aqueous extracts from tangerine (Citrus reticulata L. var. Clementina) peels as a sustainable botanical insecticide. Phytochemical screening, corroborated by Fourier-Transformed Infrared (FTIR) and Reversed-Phase High-Performance Liquid Chromatography (RP-HPLC) analyses, confirmed the presence of a rich profile of secondary metabolites, notably phenols, flavonoids, and steroids, with a predominant (−70%) d-limonene component. These findings are consistent with previous characterisations of citrus peel extracts [16, 37] and form the biochemical basis for the observed biological activities. The complexity of such natural extracts, often containing a synergistic blend of compounds, is a hallmark of botanical pesticides and may contribute to a broader spectrum of activity and a reduced likelihood of rapid pest resistance development compared to single-molecule synthetic pesticides [13, 38, 39].

The insecticidal efficacy of the 2.50% (v/v) formulation (F-2.50%) was particularly noteworthy, achieving reductions in foliar pest populations, such as Frankliniella occidentalis and Aphididae, by 49–62%. This level of control approached that of the conventional chemical treatments, a significant finding for a plant-derived product. The insecticidal properties of citrus extracts, particularly d-limonene, are well-documented, often attributed to neurotoxic effects, desiccation of the insect cuticle, or disruption of octopaminergic pathways [13, 40–45]. Furthermore, the observed reduction in Premnotrypes vorax larvae infestation in tubers suggests a broader efficacy that extends to soil-dwelling pests, an area requiring further investigation but highly relevant for potato cultivation, where such pests cause significant economic losses [36, 46, 47]. The antifungal activity associated with flavonoids in citrus extracts [20, 23, 24, 48] may also have contributed to overall plant health, although specific assessments of fungal diseases were beyond the scope of this study.

A critical advantage highlighted by this research is the F-2.50% formulation's minimal impact on beneficial insect populations, including Coccinellidae and Apis mellifera. This contrasts sharply with the conventional chemical treatment, which demonstrably suppressed these natural enemies and pollinators. The selectivity of many botanical insecticides, which allows for the conservation of beneficial arthropods, is a cornerstone of integrated pest management (IPM) strategies [13, 49–51]. This attribute makes the tangerine peel extract particularly suitable for agroecosystems, such as the Andean 'chakras', where biodiversity and ecosystem services, including pollination for organic honey production [10, 52–54], are highly valued. The low toxicity, further evidenced by the Caenorhabditis elegans model, where field-relevant concentrations (approximating the Eth-E-1% sample) showed minimal adverse effects, underscores the formulation's eco-friendly profile, likely due to the near-complete removal of ethyl alcohol during the pilot-scale concentration process.

Whilst the F-2.50% formulation yielded approximately 73% of the tuber production achieved with conventional chemical management, representing a difference of roughly 10 ton ha⁻¹, this outcome must be contextualised within a broader sustainability and circular bioeconomy framework [55, 56]. Several factors could partially offset the slightly lower yield. Firstly, the reduced reliance on synthetic pesticides mitigates environmental burdens, such as soil and water contamination, and slows the development of pesticide resistance in pest populations [57]. Secondly, the valorisation of agro-industrial waste (tangerine peels) aligns with biorefinery concepts, potentially creating additional revenue streams or reducing waste disposal costs, thereby improving the overall economic equation for farmers [58–60]. Thirdly, produce grown with such botanical inputs may command premium prices in organic or eco-labelled markets, compensating for yield differentials [61]. The long-term benefits of enhanced soil health and preserved biodiversity, although not directly quantified in this study, also contribute to the resilience and sustainability of the farming system.

The promising results advocate integrating tangerine peel extracts into pest management programs, particularly for smallholder farmers seeking sustainable and ecologically sound agricultural practices. Future research should focus on optimising extraction efficiency and field application protocols to bridge the yield gap alongside a more comprehensive quantitative analysis of the extract's chemical constituents and their specific modes of action. Investigating the extract's efficacy during different climatic conditions, such as the "dry" season, will also be crucial for validating its robustness across varied environmental contexts.

Comment 4: Line 45: Replace Aphis with Apis.

Response 4: Line 45: The error was corrected.

Comments 5: In the Methods section, clarify what was evaluated using FTIR and HPLC, as many readers may not be familiar with these techniques.

Response 5: To explain to the readers the purpose of using FTIR and RP-HPLC, explanations were introduced in lines 177–178 and 185–187.

Lines 177-178:

The FTIR spectrum of tangerine peel extract can help identify the main chemical functional groups present in the extract's components.

Lines 185-187:

Through the RP-HPLC spectrum obtained from the extract of tangerine, it would be possible to identify the presence and composition within the mixture of some of the main substances that constitute the extract.

Comments 6: Lines 378–380 and 395–397: These sentences should be moved to the Methods section.

Response 6: The content from lines 378-380, as you accurately suggested, has been moved to lines 258-261 in the 'Materials and Methods' section. Meanwhile, lines 395–397 have been removed, as the explanation was already provided in lines 288–293 of the 'Materials and Methods' section.

Sincerely yours,

The Authors

Reviewer 2 Report

Comments and Suggestions for Authors

The authors investigate the potential of ethanolic extracts from tangerine peel (Citrus reticulata var. Clementina) as a sustainable botanical insecticide for potato cultivation (Solanum tuberosum var. Capiro). The research combines field trials where two extract formulations, 1.25 % and 2.50 % v/v, are compared with a conventional chemical treatment and an untreated control with laboratory toxicity tests on a non-target organism, C. elegans. The main findings show that the higher concentration (2.5 %) significantly reduces pest populations (thrips, aphids, Premnotrypes vorax larvae) at levels comparable to synthetic pesticides while sparing beneficial insects such as ladybirds and pollinators. In terms of yield, using the extract achieves roughly 73 % of the production obtained under standard chemical practices. Toxicological assays reveal a moderate effect on C. elegans (LC₅₀ ≈ 6.4 %), lower than that of chemical formulations, and no acute toxicity to non-target organisms in the ecosystem (nematodes, beneficial insects). The study therefore proposes citrus-waste extract as a low-impact tool for integrated pest management, highlighting environmental benefits recycling agro-industrial waste and protecting pollinators and suggesting applications in sustainable farming systems. 

Key issues to resolve:

  1. Insufficient chemical characterisation, the dominant peak at 2.113 min (≈ 70 %) is merely attributed to d-limonene on the basis of retention time similarity, so instrumental confirmation by co-injection with a certified standard or by GC-MS is required.

  2. Agronomic impact and economic analysis, the yield obtained with the 2.5 % formulation is about 10 t ha⁻¹ lower than the conventional treatment (-27 %), therefore a cost–benefit assessment should be added, covering the production cost of the extract and the market price of the tubers;

  3. Toxicity to non-target organisms, the C. elegans assay includes ARPON in the control and the adjuvant alone reduces survival compared with buffer, so separate controls using extract without ARPON and ARPON alone are needed, and the LC₅₀ should be reported with its 95 % confidence interval together with complete statistical comparisons.

Comments on the Quality of English Language

The English is intelligible but still needs professional editing for syntax and punctuation. If possible, shorten the title to 15 words or fewer, for example, “Ethanolic Tangerine-Peel Extract as a Botanical Insecticide for Potato.”

Author Response

Dear Reviewer 2,

I hope you are doing well. Thank you very much for your thorough and detailed review of our manuscript, which will significantly enhance its quality.

Answer to Reviewer 2:

Comments 1: Insufficient chemical characterisation, the dominant peak at 2.113 min (≈ 70 %) is merely attributed to d-limonene on the basis of retention time similarity, so instrumental confirmation by co-injection with a certified standard or by GC-MS is required.

Respond 1: I agree with your viewpoint. The definitive demonstration would be to use a certified standard of d-limonene, run it through RP-HPLC under the same conditions as the extracted sample in this study, and obtain a predominant peak with a similar retention time. Furthermore, if its mass were confirmed through application in a mass spectrometer, it would be fully verified.

Unfortunately, we do not currently have access to a standard d-limonene reference, and our GC-MS is presently out of service. Therefore, we rely on the following reasoning. In our previous study, using peels of Citrus reticulata L. from the exact origin (Canton Pimampiro), but employing n-hexane and petroleum ether, we obtained an FTIR profile very similar to the one presented in this study. Additionally, it was later demonstrated via GC-MS that d-limonene was present in a peak obtained through RP-HPLC with a retention time similar to the one observed in this study, making the probability of its presence in the extract necessarily high.

Moreover, the efficacy demonstrated against various insect pests in the present study further supports this conclusion, aligning with research from multiple authors who associate such effects with the presence of d-limonene in these extracts.

Comments 2: Agronomic impact and economic analysis, the yield obtained with the 2.5 % formulation is about 10 t ha¹ lower than the conventional treatment (-27 %), therefore a cost–benefit assessment should be added, covering the production cost of the extract and the market price of the tubers;

Respond 2: The experimental data indeed demonstrate that the yields of this botanical pesticide candidate have not yet reached those of conventional chemical treatment. However, based on this single experiment, without testing other possible application schemes or formulations, it would be premature to assess the potential of this natural botanical pesticide candidate fairly.

Furthermore, the analysis should include a comprehensive evaluation of all the benefits of this candidate versus the advantages offered by conventional chemical products and their associated risks, such as soil and habitat contamination, human health concerns, threats to wild entomofauna, and the increased resistance of certain pests to chemical pesticide doses.

Recognising that this is a complex issue and that not all necessary information is currently available, we believe that further evidence is required to address it properly. Nevertheless, it will inevitably become a necessary topic of discussion when this product reaches the stage of practical commercial application.

For this reason, in this study, we limit ourselves to suggesting the possible usefulness of this pesticide candidate for small-scale Andean organic farmers, whose primary interest lies in obtaining 'chemical-free' products and who are also engaged in honey production.

Comments 3: Toxicity to non-target organisms, the C. elegans assay includes ARPON in the control and the adjuvant alone reduces survival compared with buffer, so separate controls using extract without ARPON and ARPON alone are needed, and the LC₅₀ should be reported with its 95 % confidence interval together with complete statistical comparisons.

Respond 3: Indeed, those samples were omitted (testing ARPON in its pure state and testing the extract without ARPON). However, both cases are unrelated to the application under study, and although they might have revealed interesting toxicity levels, they would never be applied; therefore, they were not analysed.

From the analysis of Figure 5a, it was determined that LC50 = 6.43 ± 0.28% (n = 3) (line 355 in the revised document).

Comments 4: The English is intelligible but still needs professional editing for syntax and punctuation. If possible, shorten the title to 15 words or fewer, for example, “Ethanolic Tangerine-Peel Extract as a Botanical Insecticide for Potato.”

Response 4: Following your suggestion to shorten the title of the paper to fewer than 15 words, the original title has been replaced by: Extract of Tangerine Peel as a Botanical Insecticide Candidate for Smallholder Potato Cultivation.

Sincerely yours,

The Authors

Reviewer 3 Report

Comments and Suggestions for Authors The manuscript presents very interesting results on the effects of tangerine peel essential oil used as a botanical pesticide as an alternative to chemical treatment for controlling fungi and insect pests in agricultural crops. Since this is precisely the objective of the research, I suggest improving the references in the introduction on the effects of this oil on insect pests, since only one article on whitefly is cited. In addition, the antifungal and antibacterial properties should also be better explored in the introduction. The possibility of using ethyl alcohol as a polar solvent is essential for lower toxicity and safety. In item 3.3 of the results, the authors present the following: "Qualitative microscopic analysis of the digestive nematode's integrity revealed notable disruptions in its structure and functionality. These findings suggest that the extract and the solvent produced a direct acute toxic effect on the nematodes' digestive system and cuticle". Regarding this result, it is necessary to present microscopic images and detail the histological analysis of the changes mentioned in the text.

Author Response

Dear Reviewer 3,

I hope you are doing well. Thank you very much for your thorough and detailed review of our manuscript, which will significantly enhance its quality.

Answer to Reviewer 3:

The manuscript presents very interesting results on the effects of tangerine peel essential oil used as a botanical pesticide as an alternative to chemical treatment for controlling fungi and insect pests in agricultural crops. Since this is precisely the objective of the research, I suggest improving the references in the introduction on the effects of this oil on insect pests, since only one article on whitefly is cited. In addition, the antifungal and antibacterial properties should also be better explored in the introduction. The possibility of using ethyl alcohol as a polar solvent is essential for lower toxicity and safety. In item 3.3 of the results, the authors present the following: "Qualitative microscopic analysis of the digestive nematode's integrity revealed notable disruptions in its structure and functionality. These findings suggest that the extract and the solvent produced a direct acute toxic effect on the nematodes' digestive system and cuticle". Regarding this result, it is necessary to present microscopic images and detail the histological analysis of the changes mentioned in the text.

(x) The English is fine and does not require any improvement.

Comments 1: ..Since this is precisely the objective of the research, I suggest improving the references in the introduction on the effects of this oil on insect pests, since only one article on whitefly is cited. In addition, the antifungal and antibacterial properties should also be better explored in the introduction…

Respond 1: A paragraph highlighting several studies demonstrating the antifungal effects of citrus peel extracts on certain pathogenic fungi has been added to the 'Introduction' section (see Lines 95-97).

Lines 95-97:

In general, many extracts and essential oils from the peels of other members of the Citrus genus have also demonstrated potential in controlling pathogenic bacteria [26–31] and fungi [32–35]. For example, C. limon and C. sinensis have shown effectiveness against potato rot disease [33].

Comments 2: In item 3.3 of the results, the authors present the following: "Qualitative microscopic analysis of the digestive nematode's integrity revealed notable disruptions in its structure and functionality. These findings suggest that the extract and the solvent produced a direct acute toxic effect on the nematodes' digestive system and cuticle". Regarding this result, it is necessary to present microscopic images and detail the histological analysis of the changes mentioned in the text.

Response 2: You are right; the image would have been highly valuable in supporting our perspective. However, due to the pigments present in the extract, the image lacked clarity, making it difficult to discern the details we mentioned. These details clearly illustrated the toxic effects of the extract at those concentrations, as reflected in the lack of nematode mobility and a specific discolouration observed upon contact with the extract. The histochemical analysis protocols are currently being refined in our laboratory for future studies involving this natural extract, whether in its present form or in other formulations to be developed. Therefore, we have labelled it as 'result not shown' (line 360).

Sincerely yours,

The Authors

Round 2

Reviewer 1 Report

Comments and Suggestions for Authors

The revised version is satisfactory, having been revised according to my previous comments, and deserves publication.

Author Response

Dear Reviewer 1,

I hope this message finds you well.

Thank you very much for your observations and comments. They have improved the quality of our manuscript.

Finally, thank you for accepting our revised manuscript and recommending its publication.

Kind regards,

Prof. Eng. José Manuel Pais-Chanfrau, PhD